# OpenReview forum: "GFMate: Empowering Graph Foundation Models with Pre-training-agnostic Test-time Prompt Tuning"
_ICLR.cc/2026/Conference — Submitted to ICLR 2026_

### Official Review · Reviewer_Fb27 · 2025-10-27

**Soundness:** 2
**Presentation:** 3
**Contribution:** 2
**Rating:** 4
**Confidence:** 5

**Summary:**

This paper proposes GFMate, a framework for pre-training-agnostic test-time prompt tuning of GFMs. The key idea is to decouple prompt optimization from pre-training, introducing two lightweight prompts that are tuned only at test time. Additionally, a Test-time Graph Complementary Learning objective leverages both few-shot labeled and unlabeled samples through complementary labeling to improve target-domain generalization. Experiments on twelve benchmark datasets claim up to 30% improvement over existing GFM prompt methods with large gains in efficiency.

**Strengths:**

1. The paper tackles an important problem: cross-domain generalization of GFMs without re-pretraining, and positions itself within an emerging research trend of test-time adaptation for graphs.

2. The method is conceptually simple and computationally light, requiring only prompt updates without retraining the GFM backbone.

3. The empirical section is extensive, covering multiple datasets, backbones, and pre-training objectives.

4. The writing is clear and the figures provide an intuitive illustration of the pre-training-entanglement problem and the proposed test-time workflow.

**Weaknesses:**

1. Conceptual novelty is overstated. The claimed “pre-training-agnostic” feature is somewhat misleading: most existing few-shot or prompt-based GFM methods (e.g., SAMGPT, MDGPT) also freeze the backbone and do not require coupling with pre-training. The difference between few-shot fine-tuning and test-time tuning is incremental.

2. The paper repeatedly asserts that prior prompts are “pre-training-entangled” but does not empirically demonstrate that this entanglement actually harms transferability.

3. The theoretical contribution (the “Excess Risk Bound”) offers no genuine insight into the TGCL mechanism. It restates a generic Rademacher complexity bound without linking assumptions to the graph setting or validating them empirically.

4. The role of complementary labeling is weakly justified. The entropy-based pivot-layer strategy is heuristic, and there is no analysis comparing it with pseudo-labeling or entropy minimization baselines.

5. Some comparisons seem selective: BRIDGE (ICML 2025), a closely related multi-domain GFM with generalization guarantees, is discussed in related work but not included in experiments, which undermines the completeness of evaluation.

6. The datasets used are small-scale academic benchmarks (Cora, Citeseer, Texas, etc.), which makes it unclear whether the approach scales to true “foundation” settings.

**Questions:**

1. What concrete evidence shows that “pre-training entanglement” limits cross-domain generalization? Could you provide an ablation where pre-training and prompt learning are decoupled within SAMGPT or MDGPT for comparison?

2. How sensitive is the TGCL loss to noisy complementary labels? Have you compared it with simple entropy minimization or self-training?

3. Can the proposed method handle real large-scale GFMs (e.g., GraphMAE, OpenGraph) or text-attributed graphs, or is it limited to small GNN-based backbones?

4. Since GFMate tunes prompts on unlabeled test data, how do you avoid potential label leakage or overfitting to test distribution shifts?

---

> ### Author Response · Authors · 2025-11-19
> **Response to Weakness 1**
>
> **We sincerely appreciate the time you've spent on our paper and the constructive questions and concerns you've provided, which are greatly helpful for us to improve our paper.** We have thoroughly addressed your questions. We summarise them and **highlight** the key information for improved clarity and readability as follows:
>
> ### **Re W1 (1): Clarification for Pre-training-agnostic Prompt**:
>
> > Reviewer: “The claimed “pre-training-agnostic” feature is somewhat misleading: most existing few-shot or prompt-based GFM methods (e.g., SAMGPT, MDGPT) also freeze the backbone and do not require coupling with pre-training.”
>
> We would like to kindly clarify that all our claims of **the pre-training agnostic concept focus on the prompt tuning, but not the backbone** tuning, as **most GFMs do not need to fine-tune the backbone**. The prompts in the existing and mentioned GFMs, SAMGPT [1], MDGPT [2] and MDGFM [3], are inherently coupled with the pre-training process.
>
> For example, in SAMGPT, the domain structural prompts are injected into the model message aggregation process and jointly optimised with the model through graph contrastive learning. Similarly, in MDGPT, the domain tokens are injected into the feature and are also pre-trained together with the model under a link prediction objective. **These pre-trained prompts are indeed coupled with the pre-training**.
>
> ### **Re W1 (2): Difference between few-shot and test-time prompt tuning tasks**:
>
> > Reviewer: "The difference between few-shot fine-tuning and test-time tuning is incremental."
>
> Our proposed test-time graph prompt tuning is fundamentally different from the existing few-shot prompt tuning task as discussed in Remark 2 (line 121).
>
> This distinction primarily lies in how the abundant testing data is utilised, specifically:
>
> ### **Existing Few-Shot Prompt Tuning GFM Methods (e.g., SAMGPT [1], MDGPT [2] and MDGFM [3]):**
> * They operate in a transductive setting where abundant testing data is accessible.
>     * Note that this testing information is only used **passively** as neighbourhood context when encoding the few labelled nodes as detailed in Remark 2 line 121.
> * The substantial distribution gap between the scarce labelled samples and the abundant unlabelled testing samples is left unaddressed.
>
> ### **Our Test-Time Prompt Tuning for GFMs:**
> * It **actively makes use of the accessible testing samples**, which previous methods overlook.
> * It uses target domain data to **directly optimise the prompts during inference, with no reliance on prompt pre-training and model pre-training strategies**
> * This active use of testing data enables the prompts to adapt more effectively to the testing data in the unseen target domain, leading to improved generalisation of the GFM on the unseen target domain.
>
> We have added a new Remark 4 in line 202 to address this important concern. **Also, a detailed discussion between the pros and cons of the existing GFM prompt and our method is added in Appendix H.**
>
> [1] Xingtong Yu, Zechuan Gong, Chang Zhou, Yuan Fang, and Hui Zhang. Samgpt: Text-free graph foundation model for multi-domain pre-training and cross-domain adaptation. In WWW, 2025.
>
> [2] Xingtong Yu, Chang Zhou, Yuan Fang, and Xinming Zhang. Text-free multi-domain graph pretraining: Toward graph foundation models.
>
> [3] Shuo Wang, Bokui Wang, Zhixiang Shen, Boyan Deng, and Zhao Kang. Multi-domain graph foundation models: Robust knowledge transfer via topology alignment. In ICML 2025.

---

> ### Author Response · Authors · 2025-11-19
> **Response to Weakness 2 and Question 1**
>
> ### **Re W2 & Q1 (1): Pre-training-entangled prompts limit generalisation in GFM**:
>
> > Reviewer: “What concrete evidence shows that “pre-training entanglement” limits cross-domain generalization?”
>
> We empirically demonstrate that existing GFM methods with pre-training entangled prompts limit GFM transferability in **Section 2 (line 126).** The logic behind is that:
>
> 1. In Figure 2, we observe that the existing pre-training entangled prompt method, represented by SAMGPT, fails to account for data-level variation in the target domain because **different target domain graphs exhibit distinct node neighbourhood patterns, where such target domain patterns cannot be effectively captured by the pre-training-entangled domain prompts that are pre-trained on the source domain in SAMGPT.**
>
> 2. In Figure 3, we empirically demonstrate that SAMGPT with pre-trained prompt on the source domain also **fails to account for the distribution shift that exists in the target domain by fine-tuning on few-shot samples.** This is because in GFM cross-domain scenarios, the target domain is unseen and can have a different data distribution from pre-training source domains. Therefore, the pre-trained prompt in the existing method cannot easily be generalised to the target domain if it is not similar or within the pre-training set. Further examples on MDGPT are discussed in the response to W2&Q1 (2) below.
>
> Together, these empirical observations motivate our design of pre-training agnostic test-time prompts for better cross-domain generalisability of GFMs.
>
>
>
> ### **Re W2&Q1 (2): Ablation of Decoupling Pre-training and Prompt**:
>
> > Reviewer: “Could you provide an ablation where pre-training and prompt learning are decoupled within SAMGPT or MDGPT for comparison?”
>
> The prompts in SAMGPT and MDGPT are pre-trained alongside the models and cannot be fully decoupled. If prompt learning were decoupled from pre-training, their methods would perform even worse due to the absence of pre-trained prompts, as their method design fundamentally depends on such pre-training-entangled domain prompts on the source domain.
>
> * Taking MDGPT as an example, the mixing prompt in MDGPT contains domain tokens that are pre-trained on the source domains, and this prompt prioritises knowledge specific to certain source domains, as stated in their paper. Although MDGPT aim to adapt to the target domain by aggregating the pre-trained domain tokens, **it generally assumes that the target domain is closely related to the source domain** and can be represented by a combination of the source domain prompts, as evident in their paper **Section 4.3 on page 5.** Therefore, they coupled their prompt with pre-training domains.
>
> However, **in the GFM cross-domain setting, the target domains are unseen and may differ significantly from the pre-training source domains**, which makes the target domain not easily to be represented by a linear combination of the fixed pre-trained domain tokens. This observation also holds for SAMGPT and MDGFM, where the prompts, or parts of them, are derived from the pre-training stage on the source domain set.
>
> In conclusion, **their prompt designs are coupled with pre-training as they rely on the assumption that the target domain is closely related to pre-training source domains.**

---

> ### Author Response · Authors · 2025-11-19
> **Response to Weakness 3**
>
> ### **Re W3: Theoretical insights**:
>
> > Reviewer: “... the "Excess Risk Bound"... offers no genuine insight into the TGCL mechanism ... without linking assumptions to the graph setting or validating them empirically.”
>
> We apologise for not making the theoretical insights sufficiently clear in the main text. **A detailed discussion is originally in Appendix D**, where the excess risk of the test time learning loss is upper bounded by a term that increases with the number of classes and decreases at a rate of $O(1/\sqrt N)$, with $N$ denoting the number of testing samples. Therefore, the following **two interpretations with empirical validations** are made:
>
> 1. Using **more unlabelled samples will lead to a smaller excess risk**. This corresponds to the main design motivation of GFMate to develop the test-time prompt tuning to utilise the unlabelled test samples to improve the GFM model. This can be **verified by the experiment in Table 10**, where **different amount of unlabelled data is used** for test-time prompt tuning. The conclusion is **the more the better**. This is updated and highlighted in **Appendix E.4.**
>
> 2. If there are **less classes, the excess risk will be smaller**. This corresponds to your suggestion in the comparison of binary classification and multi-class scenario. **The experiment in W1 above also verifies this theoretical intuition. We have added it in Table 9 and Appendix E.3.**
>
> **We appreciate your comment and have added these insights into our revised paper, line 330 and highlighted them. Also, we added the empirical experiments in Appendix E.3 and E.4 to validate and link to the above two theoretical insights.**

---

> ### Author Response · Authors · 2025-11-19
> **Response to Weakness 4 and Question 2**
>
> ### **Re W4 (1): Justification of the role of complementary labelling and noisy complementary labels**:
>
> > Reviewer: "The role of complementary labelling is weakly justified."
>
> The role of complementary labelling is to provide reliable tuning signals to guide the centroids to move toward a desired direction. This is justified and motivated by the observation that **directly using pseudo labels leads to severe degradation due to inaccuracy, as discussed in Section 3.4 line 287 and Appendix E.6.**
>
> As the test-time complementary labels are predicted instead of ground-truth, it may contain noisy labels. Therefore, in Appendix E.7, we evaluate the accuracy of the predicted complementary labels compared with simple pseudo labels. The higher accuracy of our complementary labels demonstrates the important role of complementary labelling in GFMate.
>
> ### **Re W4 (2): Motivation for entropy-based pivot layer selection**:
>
> > Reviewer: "The entropy-based pivot-layer strategy is heuristic"
>
> Our proposed entropy-based pivot layer strategy is **well-motivated by the empirical observation** that **prediction performance varies across layers for target domains**, as discussed in the paper **(line 287) and the observations in Figure 2.** Directly predicting the complementary labels with a fixed layer can result in suboptimal performance on an unseen target domain. Therefore, we **need to dynamically sample the layer** with the top confidence during the test time prompt tuning process to **achieve a more stable transfer to each target domain**.
>
> ### **Re W4 & Q2 (3): Comparison with pseudo label self-training**:
>
> > Reviewer: "... no analysis comparing it with pseudo-labeling ... or ... self-training."
>
> We have validated the effectiveness of TGCL by comparing the noisy pseudo-complementary labels predicted by the pivot layer with those from the fixed output layer, as shown in **Table 11 in Appendix E.7.** The results show that **GFMate with complementary labels from the entropy-based pivot layer strategies obtains more accurate complementary labels than the fixed output layer.**
>
> **We do have compared GFMate with pivot layer strategies with pseudo label prompt tuning in Appendix E.6,** as shown in Table 10. By comparing GFMate with test-time prompt tuning on different ratios of pseudo-labelled testing samples, we verify that GFMate with pivot layer-based TGCL achieves the best performance across all datasets and all shot settings. We have highlighted this important section in the revised paper on page 24.
>
> ### **Re W4 & Q2 (4): Comparison with self-training by entropy minimisation**:
>
> > Reviewer: "There is no analysis comparing it with ... entropy minimization baselines... how it compares with simple entropy minimization ..."
>
> **We follow your suggestion and further compare GFMate with pivot-layer complementary learning combined with test-time self-training via entropy minimisation.** In this setup, GFMate is encouraged to produce more confident predictions rather than relying on any complementary labels. The results are as follows:
>
> | |Methods|Chameleon|Cora|Citeseer|Photo|
> |-|-|-|-|-|-|
> | |GFMate-Entropy|39.46±3.74|48.52±7.09|33.98±7.84|55.60±2.96|
> | |**GFMate-PivotComp (ours)**|47.25±6.11|59.68±5.37|56.25±13.33|58.85±2.17|
>
> It is clear that GFMate with pivot-layer complementary labelling strategies significantly outperforms entropy minimisation training methods. **The results from the pseudo-label self-training and entropy minimisation self-training highlight the crucial role of complementary learning in GFMate.**

---

> ### Author Response · Authors · 2025-11-19
> **Response to Weakness 5**
>
> ### **Re W5: Adding comparison with BRIDGE**:
>
> > Reviewer: “...Some comparisons seem selective: BRIDGE...”
>
> We sincerely apologise for missing this important baseline and adding BRIDGE to our comparison. We **follow their open-sourced code and conduct a fair comparison in a consistent GFM setting data split** for training and testing as specified in our implementation section. We carefully tune their parameters following the instructions in their GitHub and paper. Below, we provide the one-shot node classification:
>
> | |Methods|Chameleon|Cora|Citeseer|Squirrel|
> |-|-|-|-|-|-|
> | |SAMGPT|38.12±8.90|52.83±12.04|47.76±10.55|25.75±6.29|
> | |BRIDGE|32.75±6.62|44.08±9.54|38.89±8.72|19.89±9.02|
> | |**GFMate**|47.25±6.11|59.68±5.37|56.25±13.33|27.02±6.22|
>
> The results show that although **BRIDGE demonstrates a competitive performance, GFMate can still outperform BRIDGE**. This is mainly due to our effective test time prompt tuning paradigm, which explicitly leverages the test data and enables better adaptation to the unseen target domain, whereas BRIDGE does not utilise such test time information.
>
> **We follow your suggestion and have included BRIDGE as our twenty-first baseline in Table 3 in our revised paper and we have updated its results into our main Table 1.**

---

> ### Author Response · Authors · 2025-11-19
> **Response to Weakness 6**
>
> ### **Re W6: Large scale dataset**:
>
> > Reviewer: “The datasets used are small-scale academic benchmarks.”
>
> We fully agree that using large-scale datasets is important for evaluating the foundation setting of GFMs. For this reason, we (1) **follow SOTA GFM baselines [1, 2]** for the dataset choice, and (2) **already include large-scale datasets** such as Arxiv with 169,343 nodes and 1,166,243 edges, Amazon-Photo with 238,162 edges and Squirrel with 217,073 edges, as reported in Table 4. Our setting covers a wide range of datasets with 9 node classification datasets and 3 graph classification datasets in order to **assess the generalisability of GFMate across domains with different data scales.**
>
> To be noticed, on **large-scale datasets Arxiv, most existing GFMs encounter out-of-memory** issues in Table 1, whereas **GFMate can scale to the large-scale situation** and achieve clear performance gains.
>
> [1] Haihong Zhao, Aochuan Chen, Xiangguo Sun, Hong Cheng, and Jia Li. All in one and one for all: A simple yet effective method towards cross-domain graph pretraining. In SIGKDD, 2024a.
>
> [2] Xingtong Yu, Zechuan Gong, Chang Zhou, Yuan Fang, and Hui Zhang. Samgpt: Text-free graph foundation model for multi-domain pre-training and cross-domain adaptation. In WWW, 2025.

---

> ### Author Response · Authors · 2025-11-19
> **Response to Question 3**
>
> ### **Re Q3: Large-scale GFMs and handling of text-attribute graphs**:
>
> > Reviewer: “Can the proposed method handle real large-scale GFMs (e.g., GraphMAE, OpenGraph) or text-attributed graphs, or is it limited to small GNN-based backbones? “
>
> **Thank you for your thought-provoking question. We address each of your concerns one by one as follows.**
>
> ### (1) **Large Scale GFM**
>
> GFMate can apply to large scale GFMs. If you refer to large-scale GFM as GraphMAE, which is GNN encoding with masked autoencoding based self-supervised learning, and OpenGraph, which is LLM augmented data with graph transformer encoding, **our GFMate can indeed handle these pre-trained large-scale GFMs** given that the prompts in GFMate are pre-training agnostic and can be easily integrated with different pre-trained GFMs. We also **did this experiment by incorporating our prompt design with other GFM in a plug-in style in Section 4.4 and Table 3 (highlighted in line 477)**, which verifies the flexibility and generalisability of GFMate. Therefore, GFMate is not limited to the small GNN backbones.
>
> ### (2) **Text-attributed Graphs**
>
> **GFMate can also handle text-attribute graphs** by employing a text encoder, such as Bert, to encode the text into continuous features, as exemplified by the Cora and Arxiv datasets, which are originally text-attributed graphs.
>
> ### (3) **Backbone Choice**
>
> **We respectively clarify that our backbone choice of GFMate on GNN-based GFM for experiments is not limited and has broader application than other backbones, such as LLM-based GFM.** As discussed in Appendix C.2, LLM backboned GFM methods utilise textual information in graphs by harnessing the language modelling capabilities of LLMs.
>
> Specifically, LLM-based GFM methods convert the graph learning task into a language understanding task through the testing attributes, which gives them a fundamentally different technical scope from our graph-based setting. Despite the large number of parameters in their backbone LLM, **these backbones are inherently restricted to text-attribute graphs and cannot be applied to broader applications or more general graph types without text attributes [1, 2]**, such as the biological dataset ( e.g., PROTEINS, COX2, BZR) in our paper. In contrast, GNN backboned GFMs operate in continuous feature spaces and **have a broader application** to general text-free graphs, such as the PROTEINS datasets in our Table 19.
>
> To be noticed, **GFMate make no prior assumption about the specific backbone choice as our prompt design is model agnostic.** Therefore, our method **is not limited to the specific GNN backbone and can be plugged into any GFM methods, as in Section 4.4 and Table 3**, where we demonstrate that plugging GFMate into SAMGPT and MDGPT methods can result in significant performance improvement. Further exploration can also be carried out on other backbones that follow the LLM as encoder approach [1], where an LLM is used as a text encoder to map textual attributes into continuous representations. Even with such encoders, the test time challenges discussed in Section 2 and in the introduction still remain, and GFMate can likewise develop a pre-training agnostic prompt to address them.
>
> [1] Hao Liu, Jiarui Feng, Lecheng Kong, Ningyue Liang, Dacheng Tao, Yixin Chen, and Muhan Zhang. One for all: Towards training one graph model for all classification tasks. ICLR 2024
>
> [2] Lecheng Kong, Jiarui Feng, Hao Liu, Chengsong Huang, Jiaxin Huang, Yixin Chen, and Muhan Zhang. GOFA: A generative one-for-all model for joint graph language modeling. ICLR 2025

---

> ### Author Response · Authors · 2025-11-19
> **Response to Question 4**
>
> ### **Re Q4 (1): Discussion of label leakage**:
>
> > Reviewer: "... GFMate tunes prompts on unlabeled test data, how do you avoid potential label leakage..."
>
> * **For most few-shot prompt tuning methods in GFMs [1, 2, 3], the implementation typically adopts a transductive setting.** This allows **access to both labelled and unlabelled nodes during training**, although optimisation is performed only on the few labelled nodes, as noted in Remark 2 (Line 121). Such a transductive setup is common in graph neural networks within semi-supervised learning, where models observe both labelled nodes and unlabelled test nodes, and evaluation is conducted on the unlabelled nodes [e.g., GCN]. **Importantly, no testing labels are leaked, making this a reasonable and standard evaluation scenario consistent with existing GFMs [1, 2, 3].**
>
> * From a test-time adaptation perspective, **approaches such as test-time training [4] test-time adaptation [5] and test-time graph transformation [6, 7] also assume that models can leverage additional information from unlabelled test nodes without accessing their corresponding labels.**
>
> ### **Re Q4 (2): Discussion of test distribution shift**:
> > Reviewer: “... GFMate tunes prompts on unlabeled test data, how do you avoid ...test distribution shifts?”
>
> **We have assessed the robustness of our method under different kinds of test distribution shift in Appendix E.3. In Table 10**, we evaluate GFMate under varying ratios of test time feature shift and structure shift. The results show that GFMate remains robust under different levels of test distribution shift upon both test-time feature and structural shift. **We value your concern and have added this important part into our revised main paper in Section 4.3 and Table 2, and we have highlighted it.**
>
> ---
>
> **We sincerely thank you again for your thoughtful and constructive feedback. Your comments have genuinely helped us to improve the paper. If you have any further questions or suggestions, we would be more than happy to discuss them with you. If you would raise the score, we will be really grateful.**
>
> [1] Haihong Zhao, Aochuan Chen, Xiangguo Sun, Hong Cheng, and Jia Li. All in one and one for all: A simple yet effective method towards cross-domain graph pretraining. In SIGKDD, 2024a.
>
> [2] Xingtong Yu, Zechuan Gong, Chang Zhou, Yuan Fang, and Hui Zhang. Samgpt: Text-free graph foundation model for multi-domain pre-training and cross-domain adaptation. In WWW, 2025.
>
> [3] Shuo Wang, Bokui Wang, Zhixiang Shen, Boyan Deng, and Zhao Kang. Multi-domain graph foundation models: Robust knowledge transfer via topology alignment. In ICML 2025.
>
> [4] Wenxuan Bao, Zhichen Zeng, Zhining Liu, Hanghang Tong, and Jingrui He. Adarc: Mitigating graph structure shifts during test-time. In ICLR 2025
>
> [5] Guanzi Chen, Jiying Zhang, Xi Xiao, and Yang Li. Graphtta: Test time adaptation on graph neural networks. CoRR, abs/2208.09126, 2022.
>
> [6] Wei Jin, Tong Zhao, Jiayuan Ding, Yozen Liu, Jiliang Tang, and Neil Shah. Empowering graph representation learning with test-time graph transformation. In ICLR, 2023.
>
> [7] Mingxuan Ju, Tong Zhao,Wenhao Yu, Neil Shah, and Yanfang Ye. Graphpatcher: Mitigating degree bias for graph neural networks via test-time augmentation. In NeurIPS, 2023.

---

> ### Author Response · Authors · 2025-11-27
>
> Dear Reviewer Fb27,
>
> Thank you for taking the time to review our work.
>
> We greatly appreciate your comments and we have addressed them point by point and highlighted in the revised version. **We fully understand your busy schedule, and we remain eager to address any further questions or clarifications you may have.**
>
> Sincerely,
>
> Authors

---

> ### Comment · Reviewer_Fb27 · 2025-11-28
>
> Thank you to the authors for the detailed responses. However, due to the recent issues on OpenReview, I am currently unable to update my score.

---

> ### Author Response · Authors · 2025-11-28
>
> Thank you very much for recognising our work and responding. We would be really grateful if you can update the score later and champion our paper during the discussion.
>
> Sincerely,
>
> Authors

---

### Official Review · Reviewer_WLs1 · 2025-10-28

**Soundness:** 3
**Presentation:** 3
**Contribution:** 2
**Rating:** 4
**Confidence:** 4

**Summary:**

This paper proposes a pre-training-agnostic test-time graph prompt tuning framework, named GFMate. It consists of centroid prompt and a layer prompt to align the distribution shift between the source graph and target graph and exploit the rich neighborhood information from unlabelled node, respectively.

**Strengths:**

1. The extensive experiments demonstrate the effectiveness and efficiency of the proposed method.
2. The presentation of the proposed method is good and the paper is easy to follow.
3. This paper provides the generalization bound of test-time learning loss.

**Weaknesses:**

1. I am not fully convinced by the design of test-time graph complementary learning. The authors mention that the test-time learning loss encourages centroids to be distant from testing samples being predicted to the most dissimilar class, which might be useful when the number of classes is not large, such as 2-class or 3-class. Table 2 and Table 17 show the performance comparison in the node classification and graph classification tasks, respectively. These results in 1-shot setting show that the improvement of the proposed method over other baseline methods in binary graph classification task (e.g., around 4%) is less significant than that in the node classification task (5-8 classes classification). Intuitively, if the design of test-time graph complementary learning can remove one wrong answer (by pushing the centroid away from the most dissimilar class), the performance improvement in the binary class task can be greatly improved compared with the 5-class classification task.
2. The experimental setup is not quite clear. See question below.
3. In the introduction, the authors claim that existing GFM prompt designs are generally entangled with pre-training on source domains and cannot easily generalize to unseen target domains. This statement seems overstated, as several existing methods have incorporated mechanisms to adapt to target domains. For instance, MDGPT [1] introduces dual prompts, consisting of a unifying prompt and a mixing prompt, to enable adaptation to target domains by leveraging both unified multi-domain knowledge and tailored mixtures of domain-specific knowledge. Then, the authors assert that the domain-specific information learned in the prompts cannot be effectively transferred from source to target domains due to substantial differences in structural and feature patterns. If figure 3 is used to validate the claim, the authors should visualize both the source graph and the target graphs by GFM prompts methods **explicitly**. However, current version of figure 3 fail to convey the misalignment between test distribution and the training distribution. This makes the first challenge less important.
4. Although the paper presents a generalization bound, the authors do not provide any substantive discussion or interpretation of the theoretical analysis. It remains unclear how the derived gap contributes to understanding the model’s generalization capability, particularly under domain shift or prompt adaptation scenarios. The absence of theoretical insights, such as which factors (e.g., number of classes?) most influence the gap, makes this analysis appear superficial. Without further explanation or empirical validation linking the theoretical findings to observed performance, the inclusion of the generalization gap offers limited theoretical or practical value to the paper’s overall contribution.
5. The code is not provided.

[1] Xingtong Yu, Chang Zhou, Yuan Fang, and Xinming Zhang. Text-free multi-domain graph pretraining: Toward graph foundation models. arXiv preprint arXiv:2405.13934, 2024c.

**Questions:**

1. The experimental setup is not quite clear. In Figure 3, which graph is the GFM pretrained on? Cora? Which GFM method is used for visualization? In table 1, which graphs are used for model pretraining?
2. I am not fully convinced by the design of test-time graph complementary learning. The authors mention that the test-time learning loss encourages centroids to be distant from testing samples being predicted to the most dissimilar class, which might be useful when the number of classes is not large, such as 2-class or 3-class. Table 2 and Table 17 show the performance comparison in the node classification and graph classification tasks, respectively. These results in 1-shot setting show that the improvement of the proposed method over other baseline methods in binary graph classification task (e.g., around 4%) is less significant than that in the node classification task (5-8 classes classification). Intuitively, if the design of test-time graph complementary learning can remove one wrong answer (by pushing the centroid away from the most dissimilar class), the performance improvement in the binary class task can be greatly improved compared with the 5-class classification task.

---

> ### Author Response · Authors · 2025-11-19
> **Response to Weakness 1**
>
> **We sincerely thank your thought-provoking question and concerns, which help us to improve our paper a lot.** We have thoroughly addressed your concerns and summarise our response with **highlighted** keywords as follows:
>
> ### **Re W1 & Q2: Design of Test-time Graph Complementary Learning and Performance under Binary Classification**:
>
> > Reviewer: “... test-time graph complementary learning... might be useful when the number of classes is not large, such as 2-class or 3-class. Table 2 and Table 17 ... the improvement ... in binary graph classification task (e.g., around 4%) is less significant than that in the node classification task (5-8 classes classification). Intuitively ... the performance improvement in the binary class task can be greatly improved compared with the 5-class classification task.”
>
> Thank you for the very sharp observation and intuition. We **fully agree with you** that intuitively, TGCL is expected to achieve greater improvement in binary (2-way) classification compared to multi-class classification. Yet it could be a little bit **hard to compare the improvement across tasks (node vs graph classification) and datasets** from Table 2 and 17.
>
> But since this is a very valuable suggestion, we **conduct an experiment** to verify your intuition by **using the same dataset originally with multiple classes and grouping them into binary classes**. For a balanced classification, for Cora and Amazon-Photo, we merged four randomly selected classes into one group and the remaining three into another. Similarly, for Citeseer and Chameleon, we merged three randomly selected classes into one group and treated the remaining classes as another, thus forming a two-class setting. With the SOTA best baseline SAMGPT, the results are as follows (Bin indicates the binary situation and the number indicates the original number of classes in the datasets):
>
> | |Methods|Chameleon (5)|Chameleon (Bin)|Cora (7) |Cora (Bin)
> |-|-|-|-|-|-|
> | |SAMGPT|38.12±8.90|60.61±10.36|52.83±12.04|71.19±7.80|
> | |**GFMate**|47.25±6.11|78.88±7.56|59.68±5.37|81.77±3.51|
> | |$\Delta$|9.13|18.27|6.85|10.58|
>
> | |Methods|Citeseer (6) |Citeseer (Bin)|Photo (8) |Photo (Bin)|
> |-|-|-|-|-|-|
> | |SAMGPT|47.76±10.55|55.57±12.04|56.33±9.04|73.36±4.52|
> | |**GFMate**|56.25±13.33|69.05±8.40|58.85±2.17|84.37±1.79|
> | |$\Delta$|8.49|13.48|2.52|10.01|
>
> It is evident that (1) **GFMate** significantly **outperforms** the SOTA **baseline**; and (2) The **improvement** of GFMate over the baseline **on binary classification is larger than in multi-class classification**, as in your intuition.
>
> **We sincerely thank you again for this comment to improve the quality** of our manuscript. **We have added this to the revised paper in Table 9.**

---

> ### Author Response · Authors · 2025-11-19
> **Response to Weakness 3**
>
> ### **Re W3 (1): Clarification for Pre-training Entangled/Agnostic Prompt**:
>
> > Reviewer: "... existing GFM prompt designs are generally entangled with pre-training on source domains and cannot easily generalise to unseen target domains. This statement seems overstated, as several existing methods have incorporated mechanisms to adapt to target domains. For instance, MDGPT..."
>
> We would like to clarify that **existing GFM prompt designs are generally entangled with source domains and assume the target domain is closely related to the source domain** (e.g., MDGPT [1]). Yet in the GFM cross-domain transfer learning scenario, the target domain is unseen and may significantly differ from the source domain, thereby making the source domain pre-trained prompt hard to generalise.
>
> * **Evidence by MDGPT you mentioned**: the mixing prompt in MDGPT [1] contains domain tokens that are pre-trained on the source domains, and **this prompt prioritises knowledge specific to certain source domains, as stated in their paper.** Although MDGPT aim to adapt to the target domain by aggregating the pre-trained domain tokens, **it generally assumes that the target domain is closely related to the source domain** and can be represented by a linear combination of the source domain prompts, as evident in their paper **Section 4.3 on page 5.**
>
> However, **in the GFM cross-domain setting, the target domains are unseen and may differ significantly from the pre-training source domains**, which makes the target domain not easily to be represented by a linear combination of the fixed pre-trained domain tokens. This observation also holds for SAMGPT and MDGFM, where the prompts, or parts of them, are derived from the pre-training stage on the source domain set. Therefore, our claim holds. **We appreciate your concern and have added a detailed comparison section in Appendix G for this question.**
>
> ### **Re W3 (2): Clarification for Figure 3**:
>
> > Reviewer: "... the authors should visualise both the source graph and the target graphs by GFM prompts methods explicitly. ... Figure 3 fail to convey the misalignment between the test distribution and the training distribution..."
>
> We **fully agree with you that directly visualising the misalignment** between training distribution and testing distribution is important. Yet this is (1) **not the intention of Figure 3 to illustrate the misalignment of few-shot labels and real testing samples**, (2) **nor possible since there are no few-shot labels in pretraining** datasets to visualise. This is because GFMate and our baseline GFMs adopt a label-free pre-training strategy on the source domain to avoid label-intensive backbone training.
>
> The **original intention of Figure 3** is to say that the **few-shot labels from pre-training-entangled methods are not aligned with the testing data** from target domains, leading to potential classification error. In GFMs, the evaluation will only be on the target domains, so that Figure 3 serves as an empirical observation of how pre-training-entangled methods could have an inferior performance.
>
> If what you refer to is to look at **how the few-shot tuned GFM encodes the training data and the testing data**, the gap between these two distributions is naturally significant because in most GFM experiments, pretraining domains and testing domains are from totally different domains. This is the reason and logic behind why we proposed prompts to adapt the source-domain pre-trained model to the unseen target domain.
>
> [1] Xingtong Yu, Chang Zhou, Yuan Fang, and Xinming Zhang. Text-free multi-domain graph pretraining: Toward graph foundation models.

---

> ### Author Response · Authors · 2025-11-19
> **Response to Weakness 4 & 5**
>
> ### **Re W4: Theoretical insights on generalisation bound**:
>
> > Reviewer: “... do not provide any substantive discussion or interpretation of the theoretical analysis ... understanding the model’s generalization capability, particularly under domain shift or prompt adaptation scenarios. ... further explanation or empirical validation linking the theoretical findings ...”
>
> We apologise for not making the theoretical insights sufficiently clear in the main text. **A detailed discussion is originally in Appendix D**, where the excess risk of the test time learning loss is upper bounded by a term that increases with the number of classes and decreases at a rate of $O(1/\sqrt N)$, with $N$ denoting the number of testing samples. Therefore, the following **two interpretations with empirical validations** are made:
>
> 1. Using **more unlabelled samples will lead to a smaller excess risk**. This corresponds to the main design motivation of GFMate to develop the test-time prompt tuning to utilise the unlabelled test samples to improve the GFM model. This can be **verified by the experiment in Table 10**, where **different amount of unlabelled data is used** for test-time prompt tuning. The conclusion is **the more the better**. This is updated and highlighted in **Section E.4.**
>
> 2. If there are **less classes, the excess risk will be smaller**. This corresponds to your suggestion in the comparison of binary classification and multi-class scenario. **The experiment in W1 above also verifies this theoretical intuition. We have added it in Table 9 and Section E.3.**
>
> **We appreciate your comment and have added these insights into our revised paper, line 330 and highlighted them.**
>
> ---
>
> ### **Re W5: Code**:
>
> We greatly appreciate your interest in our work, and we will release our code repository upon paper acceptance as stated in our abstract. Since ICLR submissions will be fully public, we will open-source our model and will not risk our reputation on not following what we say in the submission.

---

> ### Author Response · Authors · 2025-11-19
> **Response to Weakness 2 and Question 1**
>
> ### **Re W2 & Q1: Experimental Setup**:
>
> > Reviewer: “In Figure 3, which graph is the GFM pretrained on? Cora? Which GFM method is used for visualization? In table 1, which graphs are used for model pretraining?”
>
> We sincerely apologise for not making the experimental setup clear enough. We summarise our response to this question as follows and highlighted these parts in the revised paper:
>
> 1. **Which Pretrained Graph?**: In Figure 3 as well as all other experiments, the models and prompts are **pre-trained with one-versus-all style** as in Line 375. It means that when we evaluate on one specific dataset, we use all other datasets to pre-train the model following GCOPE and SAMGPT. For example, the GFM in Figure 3 (a) is pretrained on all source domains except Cora, where Cora serves as the unseen target domain. We updated this explanation in the caption.
>
> 2. **Which GFM method?**: In Figure 3, we visualise the embeddings from the state-of-the-art GFM prompt method SAMGPT, which serves as a representative pre-training-entangled GFM prompt method with strong performance. We updated this explanation in the caption.
>
> 3. **Which Pretrained Graph in Table 1?**: As highlighted in our dataset section (Line 375), for all our GFM results in Table 1, the models and prompts are **pre-trained under a one-versus-all cross-domain transfer learning setting, following existing GFM works** (GCOPE, SAMGPT and MDGPT), where one dataset serves as the target domain and all remaining datasets act as the source domains. We **keep a fair comparison** by following prior work and for all GFM tuning methods.
>
> For example, in Table 1, under the Cora column, all GFMs are pre-trained on the other datasets except Cora, where Cora serves as the unseen target domain for testing. This ensures a fair and consistent comparison with other GFM methods, eliminating the need for manually selecting the pre-training domains. Moreover, in further experiments in Appendix E.2 and Table 8, we showcase an ablation study of **GFMate's effectiveness when the model is pre-trained on various source domains.**
>
> ---
>
> **We thank you again for all your constructive feedback. If you have any further questions, we would be more than happy to discuss them with you. If you would raise the score, we will be really grateful.**

---

> ### Author Response · Authors · 2025-11-27
>
> Dear Reviewer WLs1,
>
> Thank you very much for taking the time to review our work.
>
> We are very grateful for your comments and have addressed each point and highlighted corresponding revisions in the updated version. **We are happy to address any further questions or clarifications and would be glad to provide any additional information at your convenience.**
>
> Sincerely,
>
> Authors

---

### Official Review · Reviewer_pevV · 2025-10-29

**Soundness:** 3
**Presentation:** 3
**Contribution:** 3
**Rating:** 6
**Confidence:** 2

**Summary:**

The paper proposes GFMate, empowering GFMs with a pre-training-agnostic test-time graph prompt tuning framework.

**Strengths:**

1. It's important to explore GFM.
2. The paper is well motivated. It's reasonable to leverage abundant unlabeled data.
3. Centroid shifting and layer re-weighting are simple, efficient, and intuitive.

**Weaknesses:**

1. Pivot-layer selection in TGCL may be unstable under data noise or heterophily; sensitivity analysis could be stronger.
2. Assumes full transductive access to the test graph; performance under inductive or streaming settings is unclear.

**Questions:**

See weaknesses.

---

> ### Author Response · Authors · 2025-11-19
> **Response to Reviewer pevV**
>
> **We sincerely thank your insightful concerns, and we have thoroughly addressed your concerns.** We summarise our response as follows and **highlight** the keywords:
>
> ### **Re W1: Pivot-layer selection under noise and heterophily cases**:
>
> > Reviewer: “Pivot-layer selection in TGCL ... under data noise or heterophily; sensitivity analysis ...
>
> **We fully agree that assessing the sensitivity of our method to data noise and heterophily is important and can further strengthen our work.** In our design, the pivot layer selection adaptively selects the most confident layer for each unseen target domain to facilitate effective prompt tuning, **instead of relying on a fixed output layer.** This ensures the robustness of our method to the noise and heterophily cases. Accordingly:
>
> 1. **Noise** cases: In **Appendix E.3 and Table 9**, we have examined the sensitivity of GFMate with the proposed pivot layer selection to different noise ratios by randomly shuffling the feature and the structure. The results verify that GFMate with pivot layer selection **is insensitive to both feature and structure noise in the graph data** and can consistently outperform baseline GFMs, with the results from **Table 1**
>
> 2. **Heterophily** cases: Our test graph covers a large range of heterophily datasets as shown in **Table 4**, and the results on these heterophily datasets further show that **GFMate with pivot layer selection delivers strong and stable performance across both homophily and heterophily graphs.**
>
> **We appreciate your comments and have further added a robustness section for evaluating GFMate under noise cases in our main paper, Section 4.3 and Table 2.**
>
> ---
>
> ### **Re W2: Performance under Inductive and streaming setting**:
>
> > Reviewer: “Assumes full transductive access to the test graph; performance under inductive or streaming settings is unclear.”
>
> 1. The **transductive setting** in the main experiment **follows prior GFM** methods such as MDGPT and SAMGPT for a **fair and consistent, as well as a reasonable comparison**. In GFM downstream adaptation, this transductive assuption allows the few-shot tuning to utilise the graph structure in prompt tuning. Otherwise, the few-shot labelled data would very likely be individual labelled nodes instead of a connected graph. We provide **a clearer Remark 4 to clarify** this in the main paper.
>
> 2. For **inductive setting, we already included an experiment in Appendix E.6 and Table 10**, where unlabelled samples are not accessible during few-shot tuning. In such inductive cases, GFMate can still learn the pre-training-agnostic prompts by the few-shot labelled samples without using the test-time loss. It is clear that **under an inductive situation, GFMate consistently outperforms all baselines in all datasets.** This is because the prompt design in GFMate can effectively generalise to unseen target domains by few-shot learning, no matter whether unlabelled samples are being used. **We appreciate your comments and have highlighted this section in Appendix E.8.**
>
> 3. Similarly, in the **streaming setting**, where the test graph keeps changing, our GFMate would still leverage **both the few-shot labelled node** as well as the continuously **incoming unlabelled data** for prompt tuning. Also note that our test time prompt tuning is **highly efficient as shown in Section 4.2**, which makes it feasible for a streaming situation. We will leave this specifically significant exploration for future work.
>
> ---
>
> **We thank you again for your constructive feedback. If you have any further questions, we would be more than happy to discuss them with you. If you would raise the score, we will be really grateful.**

---

> ### Author Response · Authors · 2025-11-27
>
> Dear Reviewer pevV,
>
> Thank you very much for taking the time to review our work.
>
> We sincerely appreciate your comments and have responded to each point with corresponding updates highlighted in the revised version. **We are eager to address any further questions or clarifications you may have, and are happy to provide any additional information at your convenience.**
>
> Sincerely,
>
> Authors

---

### Official Review · Reviewer_ZdsV · 2025-11-03

**Soundness:** 2
**Presentation:** 3
**Contribution:** 2
**Rating:** 4
**Confidence:** 3

**Summary:**

This paper tackles two major limitations of adapting Graph Foundation Models (GFMs) to unseen target domains: (1) existing prompt tuning methods are tightly coupled with pre-training, limiting generality, and (2) few-shot supervision alone cannot capture target-domain distribution, leaving unlabeled test nodes under-utilized. To address these issues, the authors propose (1) **pre-training-agnostic test-time prompt tuning**, which decouples prompt learning from pre-training, enabling flexible adaptation, and (2) **test-time graph complementary learning**, which leverages unlabeled target nodes to mitigate distribution shift. Together, these components significantly improve GFM adaptation under scarce labels.

**Strengths:**

1. The proposed approach achieves **substantial performance gains** while maintaining superior efficiency compared to existing cross-domain GFM methods.

2. The **experimental evaluation is thorough**, covering 12 benchmark datasets with extensive comparisons, ablation studies, efficiency analyses, and compatibility tests, providing strong empirical evidence for the method’s effectiveness.

3. The paper **is clearly written and logically structured**, making the methodology and insights easy to follow.

**Weaknesses:**

1. The paper's **motivation is not sufficiently sound**. Specifically, the claim that the "pre-training–entangled" nature of existing GFM prompt designs is a significant disadvantage is not thoroughly substantiated. Intuitively, given that graph data distributions and node behaviors can vary significantly across domains, distinguishing between them during training (e.g., via domain tokens or vectors) appears to be a reasonable approach. The theoretical support for the idea that this "entanglement" is harmful is currently insufficient, which makes the core premise confusing, even in light of the strong experimental results.
2. The authors attempt to elaborate on the problem in the PRELIMINARY section, but the explanation remains unclear. It is axiomatic that node behaviors differ across domains, which necessitates the injection of domain-related information. The paper would be strengthened if the authors could emphasize **the distinction between injecting domain information at *pre-training time* versus *test-time*.** A thorough discussion of the advantages and disadvantages of each approach is needed.

3. The experimental section **lacks crucial implementation details**, making reproducibility difficult. The following key information appears to be missing:  (1) For the *Single-domain Training and Testing baselines*: The specific training tasks used and the methodology for the training, validation, and testing splits. (2) For all *GFM* baselines: The specific pre-training tasks and datasets (i.e., the source graphs) that were used.

**Questions:**

1. Why would ignoring domain information during pre-training be beneficial?
2. During pre-training, given the different data distributions of source domains, would omitting domain information lead to training instability?
3. The datasets vary significantly in size. Is it possible that a single large source domain dominates the pre-training process?
4. What is the specific meaning of "different-hop neighbourhood aggregation accuracy" in Figure 2?

---

> ### Author Response · Authors · 2025-11-19
> **Response to Weakness 1**
>
> **We sincerely thank you for your valuable question, and we have thoroughly addressed all your concerns.** We summarise our response and **highlight** the key information as follows:
>
> ### **Re W1: Motivation of Pre-training Agnostic GFM Prompts**
>
> > Reviewer: "... why the "pre-training–entangled" nature of existing GFM prompt designs is a significant disadvantage, given that ... distributions ... vary across domains, distinguishing between them during training ... appears ... reasonable ..."
>
> We would like to friendly clarify that **the main concern of a domain-related prompt** for GFM cross-domain generalisation is in **how they handle the few-shot tuning and generalise to the testing scenarios** on the unseen target domain dataset **instead of pre-training and fitting to specific domain information.** Existing pre-training-entangled prompts are generally limited in the generalisability of GFMs, the logic behind this is that:
>
> * (1) If prompts are entangled with specific pre-training domains, they **cannot easily generalise to unseen target domains when the target domains are substantially different from the source domains**. In GFM cross-domain transfer learning, the target domain is unseen and different from the source domains, as you mentioned, **graph data distributions and node behaviours vary significantly across domains**.  Therefore, the domain-specific prompt pre-trained on source domains may not contain transferable information for unseen target domains, and therefore, cannot easily generalise to these target domains. **We illustrate a more detailed example of MDGPT in the response to your Q1 below.**
>
> * (2) These pre-training-entangled prompt tuning methods are mostly **designed for their specific backbone models, and therefore not easily generalisable when applying to different pre-trained GFMs**. For example, the prompts in MDGFM are jointly pre-trained with GFMs through a specifically designed graph contrastive learning objective and cannot be directly applied to a model pre-trained with alternative strategies such as link prediction, as discussed in line 56 of the introduction. **In Section 4.4 , we validate that our designed pre-training-agnostic prompts can be applied to various backbones and GFMs, in contrast to these existing pre-training-entangled prompts that are fixed to a pre-defined backbone.**
>
> In Figure 3, we further **visualise the few-shot embeddings and the unlabelled testing embeddings** by a SOTA GFM prompt method SAMGPT, with their **pre-training entangled prompt**. It can be observed that in unseen target domains, the pre-training entangled prompts would lead to **unaligned embeddings** between the labelled few-shot samples and the unlabelled testing samples. This will introduce classification error, reflecting that the generalisation of pre-training entangled prompts from source domains to unseen target domains is not satisfactory.
>
> **We appreciate your comment and further added this important discussion to line 126 in the revised paper and highlighted it.**

---

> ### Author Response · Authors · 2025-11-19
> **Response to Weakness 2**
>
> ### **Re W2: Distinction between injecting domain information during pre-training vs test-time**:
>
> > Reviewer: "... the paper does not clearly emphasise the distinction between injecting domain information at pre-training versus test-time, nor discuss the pros and cons of each approach."
>
> We apologise for not sufficiently making this comparison clear. **We added the discussion between the existing pre-training prompt in the preliminary section in our revised paper and highlighted it.** Here we provide a detailed comparison of the advantages and disadvantages below.
> ## 1. From conceptual perspectives
>
> * **Pretraining entangled methods** in existing work prioritise knowledge specific to certain source domains and pre-train their prompt to fit the specific source domain information because they generally assume the target domain is closely related to the source domain (As specified in page 4 in [1]).
>
> * **Pretraining agnostic methods** in our work prioritise domain knowledge specific to the target domain, thereby proposing test-time prompt tuning on the target domain, eliminating the need for prompt pre-training because in GFM cross-domain scenario, the target domain is unseen from source pre-training domains and can have a different data distribution.
>
> ## 2. From technical perspectives
>
> * The prompts in **pretraining entangled methods** are trained jointly with a fixed pre-defined backbone and pretraining objective, and must be retrained whenever a new training domain appears.
>
> * The prompts in **pretraining agnostic methods** are learnt at test time, without any need for prompt pretraining and without relying on any prior assumption on the backbone model architecture or pretraining strategy. Therefore, our prompts can be plugged into various backbones and GFMs with different pre-training strategies as examined in Section 4.4 and Table 3.
>
> ## 3. Pros and Cons
>
> * For **pretraining entangled methods** in existing work, they have
>
>   1. better utilisation of source domain information when the target domain is closely related,
>
>   2. worse generalisability when the target domain differs, worse efficiency since prompts must be retrained for new domains, and worse compatibility across different GFM architectures and backbone choice.
>
> * For **pretraining agnostic methods** in our work, we have
>
>   1. better ability to exploit the testing distribution of new unseen target domains and better generalisability across different GFM backbones,
>
>   2. potential computation because of the test time prompt tuning process. In Section 4.2, we examine the efficiency of GFMate and find that it significantly outperforms existing GFMs that rely on fine-tuning, under a fair comparison at the GFM downstream adaptation stage. This is because our efficient centroid and layer prompt design does not require training a prompt for each testing sample, whereas the prompts in existing GFMs are instance-level and need to be trained separately for each node.
>
> **We fully agree with you that this discussion is important and have included it in the revised version of our main paper, as highlighted in the introduction and the new discussion section on page 3, line 126. Also, we create a new discussion section in Appendix G to include it.**
>
> [1] Xingtong Yu, Chang Zhou, Yuan Fang, and Xinming Zhang. Text-free multi-domain graph pretraining: Toward graph foundation models.

---

> ### Author Response · Authors · 2025-11-19
> **Response to Weakness 3**
>
> ### **Re W3 (1): Implementation**
>
> > Reviewer: "The experimental section lacks crucial implementation details."
>
> We sincerely apologise for not sufficiently highlighting the implementation details and we will open-source our code upon paper acceptance as stated in the Abstract. **The implementation details, including the data split, are original in Section 4 (located in the dataset and implementation subsections), and we highlighted them now in the revised paper.**
>
> We would like to firstly emphasise that, the **testing/evaluation and data split** for all methods (both single-domain methods and GFMs) are the **same and fair without any label leakage**.
>
> ### **Re W3 (2): Training, Testing and Split**:
>
> > Reviewer: "(1) For the Single-domain Training and Testing baselines: The specific training tasks used and the methodology for the training, validation, and testing splits."
>
> ### For **Single-domain** methods:
>
> * **Training**:
>
>   1. For **supervised learning (SL)** methods to train GCN, GAT, and other backbones, the few-shot samples are used to train the models using cross-entropy loss on the node classification task following GCOPE [1] and SAMGPT [2].
>
>   2. For **self-supervised learning plus fine-tuning (SSL+FT)** methods, three self-supervised learning strategies are adopted: link prediction, Deep Graph Infomax, and graph contrastive learning. The model is first trained with the self-supervised task (label-free) and then fine-tuned on the few-shot labelled samples on the same dataset.
>
>   3. For **self-supervised learning plus prompt tuning (SSL+Prompt)**, the model is first pre-trained using the self-supervised task specified in the original paper (e.g., link prediction in DAGPrompt), and then the prompt is fine-tuned using the few-shot samples. The default backbone is GCN. Details for each method are provided in Appendix A.
>
> * **Testing and Data Split**:
>
> **The remaining dataset is randomly split into a 1:9 ratio for validation and testing, following GCOPE [1].** All methods, including GFM, are trained or fine-tuned using the same few-shot labelled samples and validated and tested on the same sets, ensuring a fair comparison.
>
> ### For **GFM** cross-domain methods:
>
> * **Training**:
>
> During pre-training, a leave-one-out setting is adopted where each dataset serves as the target domain once, and all remaining datasets are used as source pre-training domains. **This ensures that all source domains are utilised without manual selection.** The backbone models are trained according to the original GFM baseline strategies.
>
> For example, when evaluating on the Cora target domain dataset, all GFMs are pre-trained on the other datasets as source domains (Texas, Cornell, Wisconsin, Chameleon, Squirrel, Arxiv-year, Citeseer, Photo). The benefit is that under this leave-one-out setting, we utilise all other domains for pre-training and do not need to manually select the source pre-training domains. This ensures a fair and consistent transfer learning setup following our baseline GFM methods GCOPE and SAMGPT.
>
> * **Testing and Data Split**:
>
> For the target domain, the validation and testing split follows the same ratio as in the single-domain setting. **This guarantees that all GFMs are evaluated fairly using consistent validation and testing data, allowing a direct comparison across methods under the same column.**
>
> [1] Haihong Zhao, Aochuan Chen, Xiangguo Sun, Hong Cheng, and Jia Li. All in one and one for all: A simple yet effective method towards cross-domain graph pretraining. In SIGKDD, 2024a.
>
> [2] Xingtong Yu, Zechuan Gong, Chang Zhou, Yuan Fang, and Hui Zhang. Samgpt: Text-free graph foundation model for multi-domain pre-training and cross-domain adaptation. In WWW, 2025.

---

> ### Author Response · Authors · 2025-11-19
> **Response to Question 1**
>
> ### **Re Q1: Clarification for not using pre-training domain information**:
>
> > Reviewer: “Why would ignoring domain information during pre-training be beneficial?”
>
> Thank you for your insightful question, we summarise **three main benefits below**. More details are in the response to your W1 and have been added to the Discussion section in line 126.
>
> ### **Benefit 1 – Better generalisation to unseen domains:**
>
> * **The main objective of a GFM is to generalise to unseen domains rather than fitting domain-specific information in the pre-training domain set.** Pre-training-entangled prompts that focus on the specific pre-training source domains may not easily generalise to unseen target domains, as **graph distributions vary across domains and the target domain may not be represented by the source domains.** This phenomenon is also observed in other foundation model research areas, such as **CV and NLP, where general information is more effective in generalisation instead of pre-training on domain-specific information**.
>
> * Example: The mixing prompt in MDGPT [1] contains domain tokens that are pre-trained on the source domains and prioritises knowledge specific to certain source domains. Although MDGPT aims to adapt to the target domain by aggregating these pre-trained domain tokens, it generally **assumes that the target domain is closely related to the source domain and can be represented by a linear combination of the source domain prompts** (as evident in the MDGPT paper, Section 4.3, Page 5). However, in the GFM cross-domain setting, where target domains are unseen and may differ significantly, this assumption often fails. A similar observation holds for SAMGPT and MDGFM, whose prompts or parts of them are derived from pre-training on the source domain set.
>
> ### **Benefit 2 – No need for prompt retraining for new domains:**
>
> * Pre-training-entangled prompts often require **retraining for each new incoming domain because they are tied to a specific source domain set.** In contrast, our pre-training-agnostic test-time prompts enable the model to adapt to unseen target domains **directly at test time without additional retraining**, improving efficiency and ease of deployment.
>
> ### **Benefit 3 – Compatibility across different models and architectures:**
>
> * Existing pre-training-entangled prompts are **tied to specific fixed backbone model architectures and pre-training strategies**, limiting their generalisability to other GFMs. For example, **the domain prompts in SAMGPT are pre-trained with a GCN using a graph contrastive task and cannot work with models pre-trained by link prediction tasks.** Pre-training-agnostic prompts remove these constraints, allowing adaptation across diverse GFM architectures and pre-training strategies.
>
> Overall, pre-training-agnostic test-time prompts improve **generalisation, efficiency, and model compatibility** in GFM cross-domain settings. **We appreciate your question and added this detailed discussion in the discussion section in line 126 and Appendix G in the revised paper.**
>
> [1] Xingtong Yu, Chang Zhou, Yuan Fang, and Xinming Zhang. Text-free multi-domain graph pretraining: Toward graph foundation models.

---

> ### Author Response · Authors · 2025-11-19
> **Response to Question 2**
>
> ### **Re Q2: Omitting Domain Information during Pre-training and Stability**:
>
> > Reviewer: “Would omitting domain information lead to training instability?”
>
> Thank you for this thought-provoking question.
>
> * If what you refer to is about **not using domain-specific information in prompt design and the stability of GFM pretraining**, we would like to clarify that **GFMate pretraining is stable**, although we do not focus on pretraining, but on prompt tuning. For our prompts, they do not need to be pre-trained. For our backbone model, the **pretraining objective is designed** as a link prediction task. We sample positive links and negative links to **enhance the pretraining stability**, which simulates the training of the general contrastive learning.
>
> * Our method's training stability can be verified by **our small variance of the results** in the main table compared with other methods. The reason why **other GFMs would have a larger variance** is partially due to their incorporation of **domain-specific prompt pre-trained on source domains**, which **need to be re-developed or fine-tuned in unseen target domains, causing the domain information in the prompt to be overwritten**. Instead, our test-time prompt is specifically designed for the target domain, avoiding any conflict between the domain information from the pre-training source domains and the unseen target domain.
>
> * If what you refer to is about **omitting some source domains during pretraining**, we have provided **a domain robustness experiment in Appendix E.3**, where we group the datasets according to their domains (as in Table 4) and omit parts of domains from the pre-training domains. For example, w/ Social means pre-train the backbone model using only social domain datasets and omit all other domains. It is evident that **GFMate maintains its effectiveness and stability** compared to baselines in Table 1, even when some pre-training domains are omitted.

---

> ### Author Response · Authors · 2025-11-19
> **Response to Question 3**
>
> ### **Re Q3: Large Source Domain in Pre-training**:
>
> > Reviewer: “Is it possible that a single large source domain dominates the pre-training process?”
>
> A single large source domain may dominate the pre-training of existing GFM prompt methods, yet **the impact on our GFMate would be largely decreased**, as **our prompts do not require and do not rely on pre-training**.
>
> During **pretraining**,
>
> 1. **existing GFMs need to pretrain both the backbone and their prompts**, which would be impacted by large-scale datasets' domination.
>
> 2. **our GFMate just needs to pretrain the backbone**, and **our prompts will not be involved until test-time prompt tuning**. When a large dataset may affect our backbone pretraining, **our test time prompts will not be affected by these source datasets**, and its only tuning on target domains will compensate for this dataset bias from backbones.
>
> In Table 11, we have evaluated the GFMate performance on the backbone that is pre-trained on different domains, where the Citation domain contains Arxiv dataset, thus it is the biggest. The results verify that GFMate can maintain its effectiveness in cases when the backbone pre-training domain is changed.
>
> **We appreciate your comment and have highlighted this important discussion in line 137 of the main text.**

---

> ### Author Response · Authors · 2025-11-19
> **Response to Question 4**
>
> ### **Re Q4: Clarification of Figure 2 Caption**:
>
> > Reviewer: “What is the specific meaning of "different-hop neighbourhood aggregation accuracy" in Figure 2?”
>
> We sincerely apologise for not stating the observations sufficiently clearly. Figure 2 illustrates that with a **fixed and pre-trained multi-layer GFM**, we directly evaluate how embeddings extracted from each layer lead to prediction accuracy on different unseen target datasets. Since **each GNN layer aggregates information from a different hop neighbourhood** and this is why we refer to this as different hop neighbourhood aggregation accuracy. We have updated the caption of Figure 2 to reflect this more clearly in **blue text**.
>
> The observations in Figure 2 show that **different target domain graphs** present distinct structural patterns, which result in substantial **performance variations across layers**. Existing GFM prompt designs, represented by SAMGPT, are **pre-training entangled** and conditioned on source domain properties, thereby **hardly capture these target domain-specific variations**.
>
>
> ---
>
>
> **We thank you again for your constructive feedback. If you have any further questions, we would be more than happy to discuss them with you. If you would raise the score, we will be really grateful.**

---

> ### Author Response · Authors · 2025-11-27
>
> Dear Reviewer ZdsV,
>
> Thank you very much for taking the time to review our work.
>
> We greatly appreciate your comments and have addressed them point by point and highlighted in the revised version. **If there remains any points that could benefit from further clarification, we would be very happy to provide additional information at your convenience.**
>
> Sincerely,
>
> Authors

---

### Author Response · Authors · 2025-11-19
**Summary of Revision**

Dear Program Chairs, Senior Area Chairs, Area Chairs, and all Reviewers,

**We thank all reviewers for their time and constructive feedback.** We have uploaded a revised version of our manuscript in response to the reviewers’ concerns. All modifications are clearly described in our detailed responses to each reviewer’s official comments. To ensure that new material is easy to identify, all added text, figures and tables are **highlighted in blue font**. For clarity, we summarise the main updates with their corresponding question below.

### For Reviewer ZdsV
* In response to w1, we explained why prompts entangled with pretraining domains limit generalisation. The added discussion is at **line 126** and supported by Figure 3.

* In response to w2 and q1, we clarified the difference between adding domain information during pretraining and at test time. The comparison is now in the introduction, at line 126, and in **Appendix G**.

* In response to w3, we added the missing implementation details in **Section 4 and Appendix A**, and clarified the leave one out pretraining setup in **Section 4 and Table 3.**

* In response to q2, we clarified training stability and referred to supporting evidence in **Section 3 and Appendix E.3.**

* In response to q3, we discussed the effect of large source domains and added the clarification at **line 137, supported by Table 11.**

* In response to q4, we clarified the meaning of **Figure 2 and updated its caption.**

### For Reviewer pevV

* In response to w1, we clarified that pivot-layer selection adaptively chooses the most confident layer, ensuring robustness under noise and heterophily. Results are highlighted in **Appendix E.3, Table 9, Table 1, and Table 4, with a robustness section added in Section 4.3 and Table 2.**

* In response to w2, we explained the transductive setting used in the main experiments (Remark 4), and reported inductive and streaming results in **Appendix E.6, Table 10, Appendix E.8, and Section 4.2.**

### For Reviewer WLs1
* In response to w1 & q2, TGCL shows a larger improvement in binary classification. Results are added in **Table 9.**

* In response to w3, pre-training-entangled prompts assume target domains are related to source domains, limiting generalisation. Details in **Appendix G.**

* In response to w4 & 1, theoretical insights show excess risk decreases with more unlabelled samples and fewer classes. **Verified in Table 10 (Section E.4) and Table 9 (Section E.3).**

* In response to w2 & q1, models are pre-trained on all source domains except the target (line 375). Figure 3 shows SAMGPT embeddings. Pre-training domain ablations are highlighted in **Appendix E.2 and Table 8.**

### For Reviewer Fb27

* In response to w1, pre-training agnostic prompts focus on decoupling prompt tuning from model pre-training. Test-time tuning actively uses target domain data, unlike few-shot fine-tuning **(Details highlighted in Remark 2 line 121, Remark 4 line 202 and Appendix H).**

* In response to w2 & q1, pre-training-entangled prompts limit cross-domain generalisation **(Figures 2–3, line 126)**. Ablation shows existing work cannot fully decouple prompts from pre-training due to dependence on source-domain knowledge.

* In response to w3, theoretical insights show excess risk decreases with more unlabelled samples and fewer classes, verified in **Tables 9–10, Appendix E.3–E.4, line 330.**

* In response to w4 & q2, complementary labelling improves accuracy over pseudo-label/self-training. Entropy-based pivot layer strategy stabilises predictions across target domains **(Details highlighted in Section 3.4 line 287, Tables 10–11 and Appendix E.6–E.7).**

* In response to w5, BRIDGE is added as an additional baseline **(Table 1, Table 3).**

* In response to w6 & q3, GFMate scales to large GFMs and text-attributed graphs using GNN or text encoders, without backbone limitation **(Details highlighted in Section 4.4, Table 3, Appendix C.2).**

* In response to q4, label leakage is avoided in the standard transductive setup, and GFMate remains robust under test distribution shifts **(Details highlighted in Section 4.3, Table 2 and Appendix E.3).**

**We hope this summary offers a clear and transparent review of our revisions and supports a more informed reassessment of our work. We will continue to refine the manuscript in line with the ongoing discussion throughout the rebuttal stage.**

Best regards,

Authors

---

### Meta-Review · Area_Chair_2986 · 2026-01-02

**Summary:**

The reviewers raised substantial concerns regarding the paper’s motivation, experimental design, and theoretical analysis. Specifically, the motivation based on the claimed “pre‑training–entanglement” of existing GFM prompt designs is not sufficiently substantiated, lacking clear justification and convincing empirical evidence. The distinction between injecting domain information at pre‑training time versus test time is also insufficiently clarified, making the advantages over existing adaptive prompt methods unclear. Moreover, important experimental details are missing, raising reproducibility concerns, and the effectiveness of the proposed test‑time learning mechanism in inductive and low‑class settings is not convincingly demonstrated. Although a generalization bound is presented, its interpretation and practical implications remain limited.

During the rebuttal phase, the authors provided additional experimental details and inductive evaluations, which help alleviate some of the reviewers’ concerns. Further theoretical clarification and empirical analysis may also partially address these issues. However, the explanation of the core motivation remains only partially convincing and would require further strengthening.

Overall, while the paper reports strong empirical results, these issues significantly weaken the clarity and conceptual contribution of the work. For these reasons, I recommend rejection.

**Reviewer Concerns:**

1. Insufficient motivation.
The motivation of the paper is considered insufficiently sound, particularly the claim that the “pre‑training–entangled” nature of existing GFM prompt designs is a major limitation. This claim lacks strong theoretical justification or convincing empirical evidence, making the core premise potentially overstated and conceptually unclear. Although the authors provided further explanation in the rebuttal, the persuasiveness of this motivation remains limited.

2. Distinction between training‑time and test‑time injection.
The paper initially did not clearly distinguish between injecting domain information during pre‑training versus at test time.
The authors provided a systematic and clear response in the rebuttal, which I believe adequately addresses this concern.

3. Insufficient experimental details.
The experimental section lacks key implementation details and open‑source code necessary for full reproducibility.
The additional details provided in the rebuttal help partially alleviate this concern; however, releasing the code during the review process would further strengthen the paper.

4. Lack inductive setting evaluation.
The lack of inductive setting evaluation was a concern in the original submission.
The authors added relevant inductive experiments in the rebuttal, which successfully resolves this issue.

5. Insufficient method design clarification.
Some claimed advantages of the proposed method lack sufficient empirical support. While the authors added experiments in the rebuttal to validate these intuitions, the underlying reason why the performance gains in multi‑class settings are less pronounced than in binary classification remains unclear, which may still raise questions for the reviewers.

6. Lack of in‑depth theoretical explanation.
The theoretical analysis initially lacked sufficient interpretation and connection to empirical results.
The authors provided more detailed explanations and validation in the rebuttal, which I believe helps resolve this concern.

**Reviewer Scores:**

For Reviewer ZdsV and WLs1, I think they will keep their score 4 unchanged.
For Reviewer pevV, I think he will also keep his score 6 unchanged.
For Reviewer Fb27, I think he will become positive and raise the score to 6.

---

### Decision · Program_Chairs · 2026-01-26

Reject